

# Novel monoclonal antibodies against house dust mite allergen Der p 21 and their application to analyze allergen extracts

Vytautas Rudokas[1], Laimis Silimavicius[1,2], Indre Kucinskaite-Kodze[1], Aiste Sliziene[1], Milda Pleckaityte[1] and Aurelija Zvirbliene[1]

[1] Institute of Biotechnology, Life Sciences Center, Vilnius University, Vilnius, Lithuania
[2] UAB Imunodiagnostika, Vilnius, Lithuania

## ABSTRACT

**Background**. Allergen extracts and recombinant allergens are used in allergy diagnostics and immunotherapy. Since allergen extracts from different manufacturers lack proper standardization regarding their composition, monoclonal antibodies (MAbs) against specific allergen components can be used for their identification and quantification in allergen extracts. This study aimed to generate MAbs against allergen Der p 21 of *Dermatophagoides pteronyssinus* for the analysis of allergen extracts.

**Methods**. Recombinant Der p 21 was expressed in *E. coli* and purified using affinity chromatography. MAbs against Der p 21 were generated using hybridoma technology. House dust mite (HDM) allergen extracts were analyzed using the newly developed sandwich enzyme-linked immunosorbent assay, Western blotting and microarray immunoassay.

**Results**. MAbs raised against recombinant Der p 21 were characterized in detail and proven to be reactive with natural Der p 21. Highly specific sandwich enzyme-linked immunosorbent assay for the quantification of Der p 21 was developed and optimized. The allergen was detected and its concentration was determined in only three of six analyzed HDM allergen extracts from different manufacturers.

**Conclusion**. HDM analysis by MAb-based immunoassays shows their differences in allergen composition. The results demonstrate the importance of allergen-specific MAbs as a tool for the characterization of allergen extracts and the need for their appropriate standardization before their use for allergy diagnostics or immunotherapy.

Corresponding author
Vytautas Rudokas,
vytautas.rudokas@gmc.vu.lt

# INTRODUCTION

An allergy is a type I hypersensitivity reaction to environmental antigens that are usually harmless to most individuals. The house dust mite (HDM) is an important and widely spread allergen source causing allergy and its related complications. The main species of HDM are *Dermatophagoides pteronyssinus* (Der p), *Dermatophagoides farinae* (Der f) and *Blomia tropicalis* (Blo t). According to the World Health Organization (WHO)

and International Union of Immunological Societies (IUIS) allergen database (http://www.allergen.org), 96 allergens from these mites have been identified, 34 of them from *D. pteronyssinus*. Group 1, 2 and 23 represent major allergens, with antigenicity 76.5%, 79.2% (*Trombone et al., 2002*) and 74% (*Weghofer et al., 2013*) respectively. Der p 21 is also an important allergen which has allergenicity of ∼25% (*Weghofer et al., 2008*; *Kidon et al., 2011*; *Pulsawat et al., 2014*; *Yuriev et al., 2023*). It is found in the midgut (epithelium, lumen and faeces) of *D. pteronyssinus* and is disseminated into the environment *via* faecal particles (*Weghofer et al., 2008*). Der p 21 is a 14.7 kDa heat-stable protein with an $\alpha$-helical secondary structure (*Weghofer et al., 2008*) and has a ∼70% and ∼40% sequence identity to group 21 allergens from dust mites *D. farinae* (*Cui et al., 2014*) and *B. tropicalis* (*Gao et al., 2007*) respectively. Der p 21 also has an average sequence identity of ∼40% with the group 5 allergens from different dust mites species (*D. pteronyssinus*, *L. destructor* and *B. tropicalis*), although no relevant serum IgE cross-reactivity has been observed (*Weghofer et al., 2008*).

Small scale and high prevalence of HDM allergens in everyday environment makes them particularly difficult to avoid. Therefore, in two of every three homes there are high enough levels of HDM allergens to be associated with an increased risk of sensitization (*Simpson et al., 2002*). Large efforts are taken to offer informative and valuable allergy diagnostic tests to cope with high number of HDM-affected patients, leading to efficient therapy (*Calderón et al., 2015*). Diagnostic tests employ HDM allergen extracts and ever-increasing number of allergen components. On the other hand, allergen-specific immunotherapy (AIT), which is the only available causal treatment form to fully eradicate HDM allergy (*Bousquet et al., 1998*), uses allergen extracts, even though AIT using recombinant allergens is being actively investigated (*Zhernov et al., 2019*). Extracts are generated directly from the allergen source, therefore they may contain many different proteins and other substances, while allergen components are individual allergen molecules that cause sensitization and allergic reactions. The problem with allergen extracts is that they lack a proper, detailed and universally accepted standardization. For example, in the United States, HDM allergen extracts are standardized based on erythema size raised by the extract adminstered intradermally to individuals highly sensitized to HDM, while in Europe, the extracts are standardized based on the wheal size determined by the skin prick testing (*Carnés et al., 2017*). Given the nature of the extracts and the vast number of conditions in their production, their comprehensive standardization is difficult to achieve. Consequently, even a greater extent of batch-to-batch variability is allowed to allergen extracts, compared to chemically derived medicinal products (*Zimmer, Bonertz & Vieths, 2017*). Currently the manufacturers of allergen extracts provide allergenic activity and protein content, while only some of them additionally provide the content of major allergens. Analysis of *D. pteronyssinus* allergen extracts from different manufacturers showed high variability of components Der p 1 and Der p 2, while some of the allergens (Der p 5, 7, 10 and 21) were not even detected (*Casset et al., 2012*). Rather than determining only allergenic activity, the solution towards producing properly standardized extracts is defining the content of individual allergen components in the extracts.

A reliable way to quantify allergen components is the use of immunoassays based on allergen-specific monoclonal antibodies (MAbs). The current study describes novel Der p 21-specific MAbs, their characterization and exploiting them for the immunodetection and analysis of Der p 21. The first part of the study aimed to express, purify and analyze the antigenicity of recombinant Der p 21 (rDer p 21). The second part was focused on generating the MAbs against rDer p 21 and applying them in the analysis of HDM allergen extracts. Portions of this text were previously published as part of a preprint (DOI: 10.22541/au.168935573.39460473/v1).

## MATERIALS & METHODS

### Production of recombinant MBP-fused Der p 21 in *E. coli*

The synthetic gene coding for Der p 21 was obtained from Invitrogen (Thermo Fisher Scientific, Waltham, MA, USA). The DNA fragment of 378 bp was digested with BamHI and XhoI restriction endonucleases and cloned into pET28-MBP-TEV vector (a gift from Zita Balklava & Thomas Wassmer, Addgene plasmid #69929; Addgene, Watertown, MA, USA) (*Currinn et al., 2016*). The resulting plasmid pET28-MBP-TEV-Der p 21 included the Der p 21 fused to the maltose binding protein (MBP) coding sequence. The synthesis of rDer p 21 with N-terminal MBP in *E. coli* was induced with 1 mM isopropyl $\beta$-D-1-thiogalactopyranoside (IPTG). After 2.5 h of cultivation at 37 °C, the cells were separated by centrifugation at 1811 x g for 15 min at 4 °C, washed with rinse buffer (20 mM TRIS, 200 mM NaCl, pH 7.4) and collected by additional centrifugation.

### Purification and cleavage of rMBP-Der p 21

*E. coli* biomass was suspended in MBP binding buffer (20 mM $NaH_2PO_4$, 200 mM NaCl, 1 mM ethylenediaminetetraacetic acid (EDTA), pH 7.4) with 1 mM phenylmethylsulfonyl fluoride (PMSF). The cells were disrupted by sonication. The soluble fraction was separated by centrifugation at 15,000 x g for 20 min at 4 °C. The rMBP-Der p 21 was purified by MBP-affinity chromatography using ÄKTA start chromatography system (Cytiva, USA) and MBPTrap™ HP one mL column (Cytiva, USA). Bound proteins were eluted with MBP elution buffer (MBP binding buffer containing 10 mM maltose). Concentration of rMBP-Der p 21 was determined with NanoDrop 2000c (Thermo Fisher Scientific, Waltham, MA, USA) spectrophotometer (extinction coefficient 80,790 $M^{-1}$ $cm^{-1}$). The fused protein rMBP-Der p 21 contained TEV protease cleavage site between the allergen and MBP. Hydrolysis reaction was performed at room temperature (RT) overnight (ON) with TEV and rMBP-Der p 21 mass ratio 1:20, adding 3 mM reduced glutathione and 0.3 mM oxidized glutathione. Since MBP and TEV had hexahistidine sequence inserted at the N-terminus, rDer p 21 was purified by immobilized metal affinity chromatography using ÄKTA start. The hydrolysis reaction mixture was buffer exchanged to Ni-NTA binding buffer (50 mM $NaH_2PO_4$, 300 mM NaCl, 10 mM imidazole, pH 8) and loaded on the HisPur™ HP one mL column (Thermo Fisher Scientific, Waltham, MA, USA). rDer p 21 was found in the sample application flow-through fractions. The concentration of rDer p 21 was measured with NanoDrop 2000c (extinction coefficient 12,950 $M^{-1}$ $cm^{-1}$) and adjusted with ImageJ software (National Institutes of Health, Bethesda, USA).

## Generation of monoclonal antibodies against rDer p 21

Three 6–8 week old female BALB/c mice were selected for immunization. Sample size was based on laboratory practice and hybridoma technology protocol, no control group, randomization or blinding were applied. Mice were immunized three times every 28 days by a subcutaneous injections with 50 µg rDer p 21, using the complete Freund's adjuvant (Thermo Fisher Scientific, Waltham, MA, USA) for the primary and incomplete Freund's adjuvant (Thermo Fisher Scientific, Waltham, MA, USA) for the secondary immunization (final volume 200 µL). Third immunization and the booster dose were injected without adjuvant. Blood samples were collected by tail bleeding before each immunization and 28 days after the last immunization. Samples were analyzed by an indirect enzyme-linked immunosorbent assay (ELISA) and a titer of rDer p 21-specific IgG was determined. Boosted dose was injected three days before the hybridization. All three immunized mice were euthanized by cervical dislocation as this method was in accordance with the requirements laid down in Annex IV to DIRECTIVE 2010/63/EU. The killing of animals was completed by confirmation of the onset of rigor mortis. No surviving mice were left at the conclusion of the experiment. The hybridization was performed as described by *Köhler & Milstein (1975)*. Briefly, the spleen cells of the immunized mouse were fused with mouse myeloma Sp2/0 cells (CRL-1581, ATCC, USA) (ratio 6:1) in the presence of polyethylene glycol solution (PEG-4000; Sigma-Aldrich, Darmstadt, Germany). Fused cells were cultivated in Dulbecco's modified Eagle's growth medium (Merck, Rahway, NJ, USA) supplemented with 15% fetal bovine serum (Merck, Rahway, NJ, USA), hypoxanthine-aminopterin-thymidine (Sigma-Aldrich, Darmstadt, Germany) and 200 µg/mL gentamicin (Carl Roth, Karlsruhe, Germany). After 2 weeks, production of rDer p 21 specific antibodies by hybridoma cells was tested by indirect ELISA. Then selected clones were cultivated and cryoconserved in liquid nitrogen.

Mice for the immunization were obtained from Vilnius University, Life Sciences Center, Institute of Biochemistry, Lithuania (Vet. Approval No LT 59–13-001, LT 60–13-001, LT 61–13-004). The permission to use BALB/c mice in the experiment was obtained from the Lithuanian State Food and Veterinary Agency (Permission No G2-117, issued 11 June 2019). Mice were supervised daily, housed under controlled conditions (temperature 22 ± 1 °C, humidity 55 ± 3%, 12 h light-dark cycle, cardboard enrichment), given constant access to standard rodent food and water *ad libitum*. The maintenance of mice and experimental procedures were performed by qualified staff and conformed to Directive 2010/63/EU requirements. No anesthesia or analgesia was given in any procedures based on hybridoma technology protocol and bioethics permission. Criteria for euthanizing mice prior to the planned end of the experiment were severe signs of distress or heavy weight loss, but no such incident has happened.

## Purification of the MAbs

MAbs were purified by Protein A affinity chromatography using ÄKTA start as previously described (*Sližiene et al., 2022*).

## Conjugation of the MAbs to horseradish peroxidase

The MAbs were conjugated with horseradish peroxidase (HRP) using the periodate method, as previously described (*Stravinskiene et al., 2019*).

## Serum specimens

A collection of blood serum specimens of allergic patients was used. Informed written consent was obtained from all involved subjects. The study was approved by Vilnius Regional Biomedical Research Ethics Committee (approval no. 158200-17-926-430). The levels of IgE specific to Der p 21 were determined by Alex$^2$ (MacroArray Diagnostics GmbH, Austria) test and ranged from 1.9 to 51.1 kU$_A$/L.

## Western blot

Sodium dodecyl sulfate-polyacrylamide gel electrophoresis (SDS-PAGE) fractionated antigen samples were transferred to a polyvinylidene difluoride membrane (Carl Roth, Karlsruhe, Germany). The membrane was blocked with phosphate buffered saline (PBS) containing 2% powdered milk. Then membrane was incubated with purified MAbs (0.2 µg/mL) or allergic patients' sera diluted 1:50 in PBS with Tween-20 (PBS-T) with 2% powdered milk for 1 h at RT. Next, the membrane was incubated with antibody conjugates (goat anti-mouse IgG (H+L)-HRP (Bio-Rad, Hercules, CA, USA, catalogue number #1721011), or mouse anti-human IgE Fc-HRP (SouthernBiotech, Birmingham, AL, USA, catalogue number #9160-05)) 1:4000 and 1:1000 dilution respectively in PBS-T with 2% powdered milk for 1 h at RT. 1-Step™ Tetramethylbenzidine (TMB)-Blotting Substrate Solution (Thermo Fisher Scientific, Waltham, MA, USA) was used and the enzymatic reaction was stopped by washing the membrane with water.

## ELISA

### Indirect ELISA

Indirect ELISA was used to select anti-Der p 21 MAbs secreting hybridomas, for determination of specificity and apparent dissociation constants (Kd) of the MAbs and reactivity of rDer p 21 with HDM-specific IgE from serum samples of patients with diagnosed HDM allergy. The 96-well plates (MaxiSorp, Thermo Fisher Scientific, Waltham, MA, USA) were coated with rDer p 21 5 µg/mL (50 µL/well) for all above-mentioned experiments or 1 µg/mL (50 µL/well) for Kd determination. The wells were blocked with PBS containing 2% bovine serum albumin (BSA). Then the plates were incubated for 1 h with purified MAbs (serial dilutions from 2 µg/mL to 33.9 pg/mL, 100 µL/well), hybridoma growth medium (50 µL/well) and allergic patients' sera (1:9 dilution, 100 µL/well) (all blood and antibody samples in ELISAs were diluted in PBS-T). After that, the plates were incubated for 1 h with antibody conjugates goat anti-mouse IgG (H+L)-HRP or mouse anti-human IgE Fc-HRP (1:5000 and 1:1000 dilution respectively, 50 µL/well). The enzymatic reaction was developed using TMB (Clinical Science Products, Mansfield, MA, USA) (50 µL/well) and stopped with 3.6% H2SO4 (25 µL/well). The apparent Kd was calculated from a titration curve and defined as a molar (M) concentration of MAb corresponding to the mid-point between the maximum OD450 value and the background.

### Competitive ELISA

The competitive ELISA was used to group the MAbs according to their recognizing epitopes. A set of 96-well plates were coated with rDer p 21 (2 µg/mL, 50 µL/well). The wells were blocked and incubated for 1 h with MAbs (20 µg/mL, 50 µL/well). Then HRP-conjugated MAbs (dilutions according to direct ELISA results, 50 µL/well) were added for 1 h. The enzymatic reaction was initiated and stopped as described above. The MAbs' competition for recognized epitopes was defined from the OD450 values in accordance with positive (no MAbs) and negative (no rDer p 21) controls.

### Sandwich ELISA

Quantification of Der p 21 was performed by sandwich ELISA. A set of 96-well plates were coated with capture MAb 4D4 (0.5 µg/mL, 100 µL/well). The wells were blocked with ROTI®block (Carl Roth, Karlsruhe, Germany). Then the plates were incubated for 1 h with allergen extract or standard antigen rDer p 21 in serial dilutions from 10–300 µg/mL for extracts; from 500 or 600 ng/mL for rDer p 21 (100 µL/well). Next, the plates were incubated for 1 h with detection conjugated MAb 1A8 (1:5,000 dilution, 100 µL/well). TMB was added (100 µL/well) followed by 3.6% $H_2SO_4$ (50 µL/well). The concentration of Der p 21 in allergen extracts was determined by fitting a logistic function.

### Inhibition ELISA

The reactivity of rDer p 21 with HDM-specific IgE was additionaly analyzed by inhibition ELISA. Blood serum sample (no. 7) of patient with diagnosed HDM allergy was analyzed with four different HDM extracts from different manufacturers (MNF #2-5). Firstly, 96-well plates were coated with rDer p 21 (1 µg/mL, 50 µL/well) and blocked with 2% BSA. Serum sample (1:20 dilution) was preincubated with HDM extract for 1 h for inhibition of Der p 21-specific IgE. The total protein amount of HDM extract from MNF #2 was chosen for preincubation, based on sandwich ELISA results, to contain 2:1 mass ratio of nDer p 21 in the extract (100 ng) *vs* rDer p 21 in a plate well (50 ng). For HDM extracts from MNF #3-5 containing lower or undetectable amounts of Der p 21, the total protein amount for preincubation was chosen to be the same as for the extract from MNF #2. After the blocking step, the preincubated samples were added to the wells (100 µL/well) and incubated for 1 h, then the conjugate anti-human IgE Fc-HRP (1:1,000 dilution, 50 µL/well) was added and incubated for 1 h. The enzymatic reaction was developed and stopped as described above. The inhibition efficiency was calculated from the OD450 values of the positive control (serum preincubation with purified rDer p 21) and negative control (serum preincubation without HDM extract).

## Microarray immunoassay

To analyze rDer p 21 antigenic properties using serum samples and to determine the specificity of the MAbs, rDer p 21, thermally denatured rDer p 21 and rMBP-Der p 21 (0.3 mg/mL) were printed on 2D-Epoxy glass slides (PolyAn GmbH, Berlin, Germany). For determination of antibody reactivity with rDer p 21 and Der p 21 in HDM extracts, a serially diluted rDer p 21 with a concentration ranging from 100 to 0.00305 µg/mL and six different HDM extracts diluted 16 times from their stock concentrations were printed

on glass slides. Human serum albumin, rMBP, HRP and PBS were printed as negative controls. Each allergen and control were printed as a single droplet ($\sim$450 pL/drop) in three replicates onto glass slides using sciFLEXARRAYER SX microarray printer (SCIENION GmbH, Berlin, Germany). The slides were blocked with PBS-T containing 2% BSA (blocking buffer) for 30 min at RT. After that, the slides were incubated for 2 h with either serum samples (1:4 dilution with blocking buffer, 80 µL/well) or MAbs (0.5 µg/mL for specificity analysis and 0.2 µg/mL for reactivity of the MAbs analysis, diluted with blocking buffer, 80 µL/well). Then the slides were incubated for 30 min with the mouse anti-human IgE Alexa Fluor® 647 (SouthernBiotech, USA, catalogue number #9160-31) or goat anti-mouse IgG Fc Alexa Fluor® 647 (SouthernBiotech, USA, catalogue number #1033-31) (1 µg/mL, diluted with blocking buffer, 80 µL/well). The slides were scanned using InnoScan 710 AL microarray scanner (Innopsys, France). The images were analyzed with MAPIX software (Innopsys, France).

## Statistical analysis

The antigenicity analysis data was evaluated by the GraphPad Prism software (Dotmatics, Boston, MA, USA). The results were presented as mean $\pm$ standard deviation (SD), $n = 3$. The results of microarray immunoassay experiments were presented as average mean $\pm$ SD of median fluorescence intensity (MFI) values, $n = 3$. The correlation between results obtained by Alex$^2$ and ELISA, and between Alex$^2$ and microarray immunoassay was determined by calculating the Pearson correlation coefficient using the GraphPad Prism software. Calculations of the apparent Kd were performed using the OriginPro 8 software (OriginLab, Northampton, MA, USA), and the results were presented as mean $\pm$ standard error of mean (SEM), $n = 3$. The standard curve parameters R$^2$ and the limit of detection for the optimized sandwich ELISA were calculated using OriginPro 8 software and according to (*Armbruster & Pry, 2008*), respectively. Calculations of Der p 21 concentration in allergen extracts were performed using the SoftMax® Pro software (Molecular Devices, San Jose, CA, USA), and the results were presented as mean $\pm$ SD, $n = 3$.

## RESULTS

### Expression, purification and analysis of rDer p 21

rMBP-Der p 21 (calculated molecular weight of 58.79 kDa) was produced in *E. coli* and was found in a soluble fraction of the cell lysate. rDer p 21 was purified by 2-step affinity chromatography (Fig. S1). Firstly, rMBP-Der p 21 was purified by MBP-specific affinity chromatography. The yield of rMBP-Der p 21 was about 1.6 mg from 1 g of wet *E. coli* biomass. To detach MBP from Der p 21, different conditions of the cleavage by TEV protease were analyzed: TEV and rMBP-Der p 21 mass ratio 1:50 and 1:20; temperatures 4 °C and RT. The mass ratio 1:20 and temperature 4 °C were found to be optimal for the cleavage. After hydrolysis, the second purification step by Ni-affinity chromatography was applied. The rDer p 21 (14.74 kDa), which has no hexahistidine tag, was found in the flow-through fractions and the MBP was found in the eluted fractions. The yield of the purified rDer p 21 was 0.34 mg from 1 g of wet biomass. However, a small amount
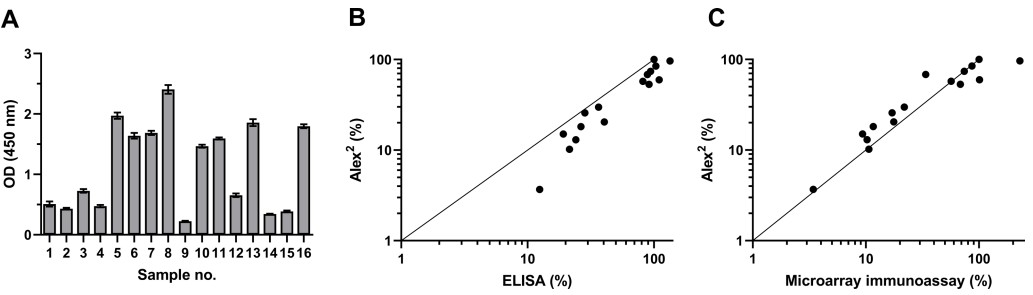

**Figure 1 rDer p 21 antigenicity analysis by ELISA and microarray immunoassay using serum samples of 16 HDM allergic patients.** (A) rDer p 21-specific IgE determined by ELISA ($n = 3$, mean $\pm$ SD). (B) Correlation between results obtained by Alex$^2$ and ELISA, (C) correlation between results obtained by Alex$^2$ and microarray immunoassay ($n = 3$). For correlation analysis (B and C), the values obtained from different tests (Alex$^2$, ELISA and microarray immunoassay) were converted to percentages relative to the serum with the highest kU$_A$/L value determined by the Alex$^2$ test being set at 100%.

of MBP was left in the final sample of the purified rDer p 21, possibly due to its partial degradation and loss of the hexahistidine tag. According to calculations using the ImageJ software, the amount of MBP in the rDer p 21 sample was 20.7% (Fig. S1C).

To investigate the antigenic properties of the purified rDer p 21, its reactivity with blood serum specimens of patients with diagnosed HDM allergy was analyzed by an indirect ELISA and the microarray immunoassay. The sera were previously tested by the commercial multiplex test Alex$^2$ on its different allergen components, including rDer p 21. The results of ELISA and microarray immunoassay based on the use of our rDer p 21 were compared with those obtained in the allergy diagnostics platform Alex$^2$. The purified rDer p 21 was recognized by HDM-specific IgE both in ELISA and the microarray test (Fig. 1) thus confirming the proper antigenic structure of the purified rDer p 21. High correlation between Alex$^2$ results with the in-house rDer p 21-based tests—ELISA and microarray immunoassay—was observed with the values of Pearson correlation coefficient 0.9470 ($p < 0.0000001$) and 0.8220 ($p < 0.0001$), respectively (obtained values from different tests were converted to percentages in accordance with the serum with highest kU$_A$/L value (Alex$^2$ test) being 100%).

## Generation and characterization of anti-Der p 21 MAbs

After fusion of spleen cells of the immunized mouse with myeloma cells, hybrid clones were screened on rMBP-Der p 21 and then retested both on rDer p 21 and MBP (negative control) to exclude their reactivity against MBP. In total, five hybridoma clones producing rDer p 21-specific MAbs were generated (Table 1). Isotype-specific ELISA showed that the MAbs are of IgG1 subtype ($\kappa$ light chain). The affinity of MAbs was analyzed by measuring the apparent Kd by indirect ELISA ($n = 3$). The Kd ranged from 61.1 $\pm$ 3.32 pM to 115.7 $\pm$ 8.26 pM, which indicates high-affinity binding.

The reactivity of the MAbs with denatured rDer p 21 and their potential cross-reactivity with irrelevant bacterial proteins was analyzed by WB (Fig. S2) using *E. coli* BL21 Star™ (DE3) lysate as a negative control. The MAbs were reactive with rDer p 21

**Table 1  Characterization of anti-Der p 21 MAbs.**

| Clone | Isotype | $K_d(pM)^*$ | Reactivity with the antigens | | |
| | | | ELISA/WB/microarray immunoassay | | |
| | | | rDer p 21 | rMBP-Der p 21 | rMBP |
|---|---|---|---|---|---|
| 1A8 | IgG1 $\kappa$ | 61.1 ± 3.32 | + | + | − |
| 4D4 | IgG1 $\kappa$ | 77.29 ± 2.9 | + | + | − |
| 2F3 | IgG1 $\kappa$ | 78.97 ± 3.23 | + | + | − |
| 2G4 | IgG1 $\kappa$ | 80.45 ± 3.48 | + | + | − |
| 8C3 | IgG1 $\kappa$ | 115.7 ± 8.26 | + | + | − |

**Notes.**

*$n = 3$, $K_d$ ± SEM.

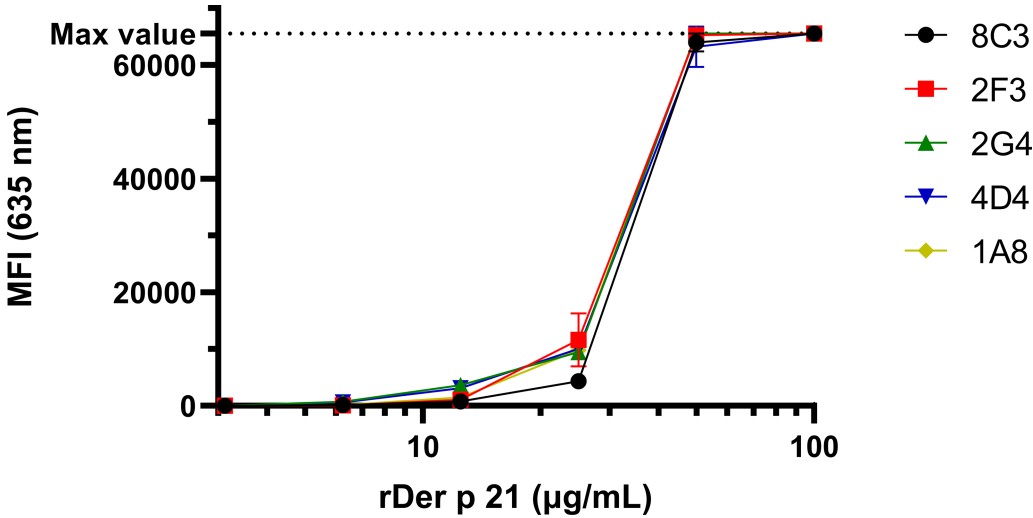

**Figure 2  The reactivity of the MAbs with rDer p 21 immobilized on a microarray glass slide.** The max detectable intensity value is 65,535. $n = 3$, mean ± SD.

and rDer p 21-MBP. No reactivity with rMBP or any other proteins from the lysate was observed. The reactivity and specificity of the MAbs were also tested by the microarray immunoassay with different antigens—rDer p 21 (Fig. 2), rMBP-Der p 21, rMBP and the denatured rDer p 21 (Fig. S3). The MAbs were reactive with rDer p 21, denatured rDer p 21 and rMBP-Der p 21, while no reactivity with rMBP used as a negative control was observed. The results indicate that all five generated hybridoma clones secrete high-affinity rDer p 21-specific MAbs, that can be applied for Der p 21 detection by different immunoassays.

## Development of sandwich ELISA for the quantification of Der p 21

To develop sandwich ELISA for Der p 21 detection, pairs of non-competing MAbs that recognize distinct Der p 21 epitopes were selected. All MAbs were conjugated with HRP and tested in a competitive ELISA. Two groups of MAbs were identified that recognize non-overlapping epitopes—group 1 (MAbs 2G4, 4D4) and group 2 (MAbs 1A8, 2F3).

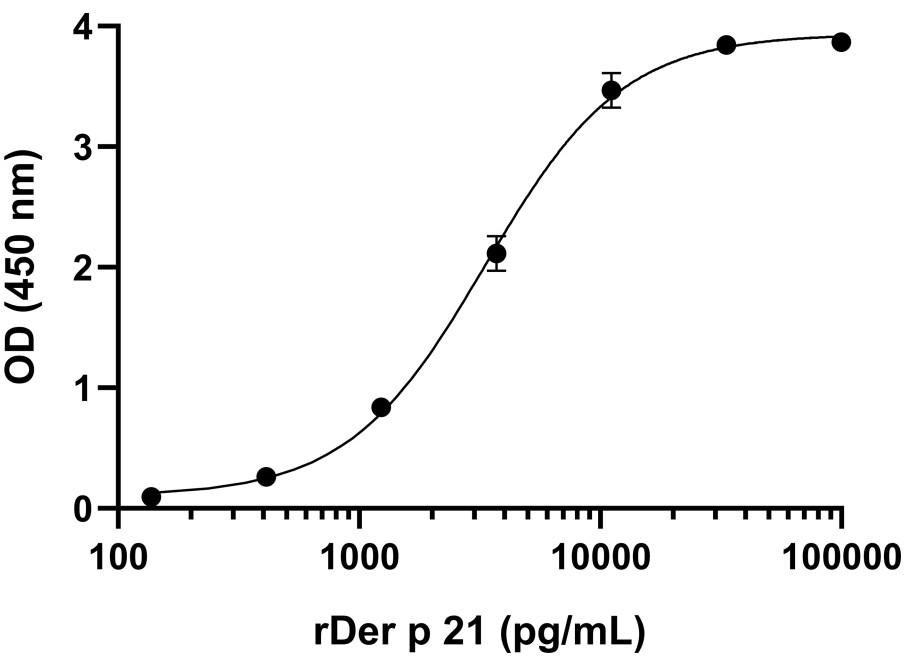

**Figure 3 Standard curve of the optimized sandwich ELISA for rDer p 21 quantification.** $n = 3$, mean $\pm$ SD.

Based on the results of the competitive ELISA, different combinations of MAbs from groups 1 and 2 were tested in sandwich ELISA format. The MAb 8C3 was excluded from testing. The highest detection sensitivity was observed using 4D4 as capture MAb, and 1A8 as detection MAb. These MAbs showed the highest affinity to rDer p 21 (Table 1). Different ELISA conditions were evaluated to optimize the assays: plate type, capture MAb 4D4 concentration, blocking solution, detection MAb 1A8 dilution. The optimized sandwich ELISA showed excellent correlation ($R^2 = 0.99973$) between rDer p 21 concentration and the OD450 value (Fig. 3) with the calculated 27.25 pg/mL limit of detection.

## Application of the MAbs for the analysis of HDM extracts

The optimized sandwich ELISA was applied to analyse six *D. pteronyssinus* allergen extracts from different manufacturers (MNF #1-6). Der p 21 was detected only in three of the analyzed allergen extracts. Its mass fraction ranged from 0.51 to 15.17 µg/mg, or mass percent from 0.05 to 1.52% respectively (Table 2). Allergen extracts were also analyzed by WB with MAb 1A8 and by microarray immunoassay with MAb 2G4 (Fig. 4). The analysis of allergen extracts with other MAbs showed similar results (Fig. S4).

The results of the microarray immunoassay (its scheme represented in Fig. S5) are in agreement with the results of sandwich ELISA, since Der p 21 was detected in the same three allergen extracts (MNF #1-3). The concentration of Der p 21 is much lower in the extract from MNF #3 (0.51 $\pm$ 0.03 µg/mg) compared to MNF #1 and #2 (15.17 $\pm$ 2.26 and 10.85 $\pm$ 0.42 µg/mg respectively). In contrast, Der p 21 was not detected in the
**Table 2  Summary of the analysis of *D. pteronyssinus* allergen extracts.** Analysis by sandwich ELISA ($n = 3$, mean $\pm$ SD) in comparison with results by Western blot and microarray test.

| Allergen extract | ELISA | | WB | Microarray immunoassay |
| | Der p 21 content | | | |
| | µg/mg | % | | |
| --- | --- | --- | --- | --- |
| MNF #1 | 15.17 ± 2.26 | 1.52 | + | + |
| MNF #2 | 10.85 ± 0.42 | 1.09 | + | + |
| MNF #3 | 0.51 ± 0.03* | 0.05 | − | + |
| MNF #4 | − | − | − | − |
| MNF #5 | − | − | − | − |
| MNF #6 | − | − | − | − |

**Notes.**

*$n = 2$.

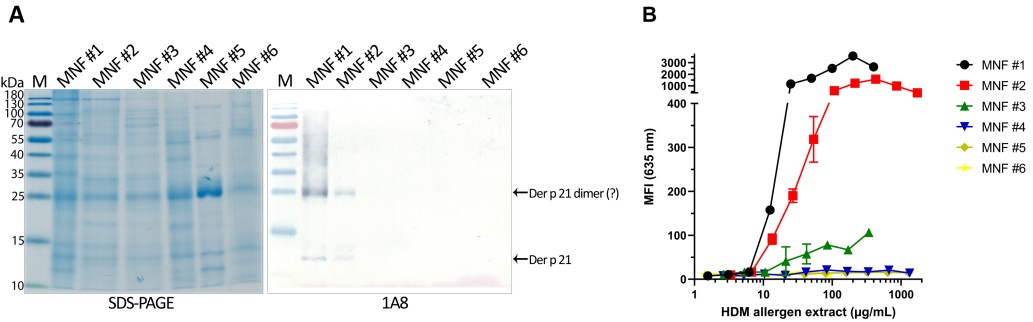

**Figure 4  Analysis of *D. pteronyssinus* allergen extracts from different manufacturers.** (A) Western blot (with MAb 1A8) and (B) microarray immunoassay (with MAb 2G4, $n = 3$, mean $\pm$ SD) analysis of *D. pteronyssinus* allergen extracts from six different manufacturers (MNF #1-6).

allergen extract from MNF #3 by WB, possibly due to lower sensitivity of this assay. Der p 21 was not detected by any of these methods in the allergen extracts from MNF #4-6.

To validate the proper folding of rDer p 21 and to confirm the results of HDM extract analysis, inhibition ELISA was performed using HDM IgE-positive blood serum sample (no. 7) and four different HDM extracts (MNF #2-5). HDM extract from MNF #2 containing the highest amount of Der p 21 caused a complete inhibition (>100%) of binding of Der p 21-specific IgE to rDer p 21, while extracts from MNF #3, #4 and #5 with lower amounts of Der p 21 caused only the partial inhibition (43, 33 and 26%, respectively) of anti-Der p 21 IgE binding to rDer p 21 (Fig. S6). These results indicate the proper folding of rDer p 21, since Der p 21-specific IgE competes with natural Der p 21 in HDM extracts for binding to rDer p 21.

## DISCUSSION

Our study describes new tools for analyzing the composition of HDM extracts and emphasizes the problem of high variability of allergen extracts, which could lead to false results of allergy diagnostic tests and unexpected AIT outcome. For example, AIT using

HDM allergen extracts induces IgG response mostly to Der p 1 and Der p 2, then, to lower extent, Der p 23 and no IgG response to Der p 5, Der p 7 and Der p 21 (*Rodríguez-Domínguez et al., 2020*). Currently, allergy molecular diagnostics are greatly advanced and offer highly detailed IgE sensitivity results that are used to tailor the patient specific recommendations and therapy suggestions. Precise allergy diagnostics are based on allergen components, while allergen extracts provide a general yes-or-no result to direct for more detailed diagnostic tests. On the contrary, AIT is primarily based on highly variable allergen extracts. Therefore, natural allergen extracts are yet to be replaced by recombinant allergens, demonstrating the importance of their detailed characterization. It is generally agreed, that standardization of allergen extracts improves both safety and efficacy of AIT. In the past, allergen extracts were only standardized for biological potency, but nowadays there is a tendency for the quantification of single allergen components in the extracts for the standardization purposes. Two main methods for the quantification of allergen components are mass spectrometry and ELISA. The first sandwich ELISA for the quantification of major HDM allergen components Der p 1 and Der f 1 was developed in 1989 (*Luczynska et al., 1989*). Even though the efforts to make reference ELISA for the quantification of allergen components proved to be unexpectedly long and challenging, two ELISA-based tests have been validated for the comparison of birch pollen and Timothy grass pollen allergen extracts in regard to their major allergens (*Zimmer et al., 2021*). The development of immunoassays for allergen quantification is mainly focused on the major allergens, but it should be noted that other components are also important for some individuals. For example, Der p 21 is a mid-tier allergen, but it binds high levels of IgE from HDM-allergic patients and has high allergenic activity (*Weghofer et al., 2008*). Consistently with other studies, we have proven that MAbs represent a highly specific tool for the quantification of allergen components, in particular for analyzing the content of Der p 21 in HDM extracts.

In the current study, we have used recombinant purified allergen (rDer p 21) for generation of Der p 21-specific MAbs. In a first step, the antigenic properties of the purified rDer p 21 were evaluated. The recombinant allergen was recognized by HDM-specific IgE from human serum samples thus confirming its antigenic similarity with native allergen. Moreover, high correlation of the in-house rDer p 21-based tests (ELISA and microarray immunoassay) with Alex$^2$ results demonstrated that rDer p 21 is a reliable diagnostic component for the molecular allergy diagnostics platforms. This result is consistent with another study demonstrating that rDer p 21 produced in *Pichia pastoris* is suitable for the *in vitro* diagnostics of HDM allergy (*Pulsawat et al., 2014*). In our study, a collection of Der p 21-specific MAbs was generated and exploited to analyze HDM allergen extracts from 6 different manufacturers by three different immunoassays—sandwich ELISA, WB and microarray immunoassay. In previous studies, the presence of many different HDM allergen components, including Der p 21, in HDM extracts has been investigated, however, the precise concentration of Der p 21 has not been determined (*Casset et al., 2012*; *Brunetto et al., 2010*). The newly developed sandwich ELISA revealed a relatively similar concentration of Der p 21 in two of the analyzed extracts produced by MNF #1 and MNF #2 - $15.17 \pm 2.26$ and $10.85 \pm 0.42$ µg/mg, respectively. The third

extract (MNF #3) contained significantly lower amount of Der p 21 - $0.51 \pm 0.03$ µg/mg. Moreover, Der p 21 was not detectable in three analyzed HDM extracts produced by MNF #4-6. These results clearly show the differences in HDM allergen composition and are in line with other studies, where the concentrations of Der p 1 and Der p 2 in different HDM extracts showed high variability, some components (Der p 5, Der p 7 and Der p 10) were poorly represented or even absent, while Der p 21 was detected in only two of ten HDM extracts from different manufacturers (*Casset et al., 2012*; *Brunetto et al., 2010*). Analysis of HDM extracts by SDS-PAGE and WB also revealed a significant variability in protein composition and suggested the dimerization of Der p 21, which is in agreement with structural studies (*Weghofer et al., 2008*). The novel microarray immunoassay was applied for Der p 21 quantification and confirmed its highest amounts in HDM extracts of MNF #1 and #2 and trace amounts in those of MNF #3, which is consistent with ELISA results. Finally, the inhibition ELISA was performed to validate the proper folding of rDer p 21 and to confirm the results of allergen extracts analysis.

In order for the newly developed sandwich ELISA to serve as a reference assay for the standardization of HDM extracts, a comprehensive analysis of both rDer p 21 as an allergen standard and the assay itself is needed. In addition to physico-chemical characterization of rDer p 21 its antigenicity analysis with a large number of human serum samples should be performed. Different synthesis and purification strategies might be employed to produce reference Der p 21. For the standardization of MAb-based ELISA, a number of allergen extracts from different manufacturers and production lots as well as spiked HDM extracts with varying concentrations of rDer p 21 have to be investigated by several independent laboratories. Only after completing thorough rounds of trials, such ELISA could be validated, certified and recommended as a standard method (*European Medicines Agency, 2008*). Then the manufacturers of HDM extracts could use this method to determine the concentration of Der p 21 in their products. Quantification of allergen components in allergen extracts has been validated with some allergens using standard ELISA methods (*Kaul et al., 2016*; *Zimmer et al., 2022*). However, it would be beneficial to implement more universal and accessible standardization tools that would enable the comparability between products from different manufacturers or the lots of each manufacturer production. Depending on the further needed Der p 21-specific degree of variability analysis, acceptance ranges of variability could be implemented. An appropriate starting point might be 50–200%. Then the individual allergen content might be included in the summary of product characteristics and its amount could be calculated per maintenance dose (*Zimmer et al., 2021*).

## CONCLUSIONS

Our study showed how generated novel antibodies against HDM allergen Der p 21 can be applied for the analysis of allergen extracts from different manufacturers. The allergen was detected in only three of six analyzed HDM allergen extracts and it shows the differences in allergen composition. This research proved that MAbs could be successfully used as a molecular tool for quantification of specific allergen in allergen extracts. The detailed

analysis of allergen extracts could be used for the standardization basis which could improve both the allergy diagnostics and AIT.

## ACKNOWLEDGEMENTS

We are grateful to Kristina Masalaite from the Institute of Biotechnology, Life Sciences Center, Vilnius University for her assistance in preparing the figure illustrating microarray immunoassay using BioRender.com and granting the license to use it in journal publication.

### Funding

This work was supported by the European Regional Development Fund (Project No. 01.2.2-LMTK-718-01-0008) under grant agreement with Research Council of Lithuania (LMTLT). The funders had no role in study design, data collection and analysis, decision to publish, or preparation of the manuscript.

### Grant Disclosures

The following grant information was disclosed by the authors:
The European Regional Development Fund under grant agreement with Research Council of Lithuania (LMTLT): 01.2.2-LMTK-718-01-0008.

### Competing Interests

Laimis Silimavicius is employed by UAB Imunodiagnostika, the Lithuanian company that specializes in the development of microarray-based assays and allergy *in vitro* diagnostics.

### Author Contributions

- Vytautas Rudokas conceived and designed the experiments, performed the experiments, analyzed the data, prepared figures and/or tables, authored or reviewed drafts of the article, and approved the final draft.
- Laimis Silimavicius performed the experiments, authored or reviewed drafts of the article, and approved the final draft.
- Indre Kucinskaite-Kodze conceived and designed the experiments, authored or reviewed drafts of the article, and approved the final draft.
- Aiste Sliziene performed the experiments, authored or reviewed drafts of the article, and approved the final draft.
- Milda Pleckaityte performed the experiments, authored or reviewed drafts of the article, and approved the final draft.
- Aurelija Zvirbliene conceived and designed the experiments, authored or reviewed drafts of the article, and approved the final draft.

### Human Ethics

The following information was supplied relating to ethical approvals (*i.e.,* approving body and any reference numbers):

A collection of blood serum specimens of allergic patients was used. Informed written consent was obtained from all involved subjects. The study was approved by Vilnius Regional Biomedical Research Ethics Committee.

### Animal Ethics

The following information was supplied relating to ethical approvals (*i.e.,* approving body and any reference numbers):

Mice for the immunization were obtained from Vilnius University, Life Sciences Center, Institute of Biochemistry, Lithuania (Vet. Approval No LT 59–13-001, LT 60–13-001, LT 61–13-004). The permission to use BALB/c mice for immunization was obtained from the Lithuanian State Food and Veterinary Agency.

### Microarray Data Deposition

The following information was supplied regarding the deposition of microarray data:

The data is available at EMBL-EBI: E-MTAB-13587, E-MTAB-13590, E-MTAB-13625.

### Data Availability

The raw data is available at Figshare: Rudokas, Vytautas (2024). Raw data. figshare. Figure. https://doi.org/10.6084/m9.figshare.24417241.v1.

### Supplemental Information

Supplemental information for this article can be found online at http://dx.doi.org/10.7717/peerj.17233#supplemental-information.

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
