# Peer review of "Novel monoclonal antibodies against house dust mite allergen Der p 21 and their application to analyze allergen extracts"

_PeerJ, doi:10.7717/peerj.17233_

## Round 0.1 · original submission · Major Revisions

Please revise the manuscript according to the referees' comments.

1. Please verify if all references are relevant to the manuscript content.
2. Please verify the statistical analysis used to validate the results.
3. All changes in the manuscript should be highlighted to make it easier for the reviewer to identify and evaluate any changes.
4. Please provide a cover letter with a point-by-point response to all reviewer's suggestions.
5. After your submission of an improved version of the manuscript, the revised version will be sent to the editors and reviewers.
6. Include an image as a brief method indicating the benefit of the fast mAbs selection and a microarray.

·

Basic reporting

No comment

Experimental design

This is a technical article: how to produce mAbs against recombinant HDM allergen Der p 21. But the product used for mice immunization/screening of hybridomas is contamined with rMBP (20%)!! Moreover, MBP has HisTag. So there is a risk that mAbs anti-Der p 21 can detect rMBP and any HisTag. It is not clear why the authors selected MBP fusion protein to express Der p 21. And particularly, because one paper described the production of rDer p 21 without fusion partner in E.coli as well.

Validity of the findings

OK. The anti-Der p 21 mAbs can detect the allergen in HDM extracts

Additional comments

1)The strategy of rDer p 21 production is weird. And particularly when the final product is pure at only 80% (20% mbp). Issues for the screening of mAbs. This contaminated preparation was used for the screening of the hybridomas producing mAbs.
2)The ALEX2 values must be provided for each human serum
3)Fig.1.B: ELISA OD %???? Difficult to understand
4)Native/denatured rDer p 21
It looks the authors got the same OD values with the monoclonals. So, all the monoclonals recognize linear and not conformational epitope?
5)To validate the folding of rDer p 21, the authors could perform competition IgE assay, using the two HDM allergen extracts apparently positive for Der p 21

·

Basic reporting

Interesting work and paper.
There are some things that need to be revised to make a really great paper.
1) show the segments of Der p 21 bound by the individual MAb.
2) make sense of the differences in binding to proteins from 6 extracts by doing inhibition with purified Der p 21 vs. Immunoblots
3) make a few minor corrections in spelling. line 230, "not Standart," Rather it is Standard. line 370, not "standartization
it is "standardization"
4) Add discussion about how THIS ELISA and MAb could be used, along with other tests, to "standardize" HDM extracts. It seems this is not fully thought out. I know the PEI and standardization (limited) from the EMA, and from FDA. I used to work in an Allergen Extract company (1980-1985). We collectively have not made sensible progress on standardization. And how much variation there could be (and to which proteins) per allergenic source. We do not make individual allergens available for testing or therapy. But we could make some real progress by the right, low-cost standardization processes. They will NOT but accepted in all countries. The companies can do most of it, but in sensible ways. Allergy doctors want better standardization. How should your system be used?

Experimental design

Generally good. BUt you need to identify sequences or "epitopes" that are detected.

Validity of the findings

Interesting but needs clarification of the parts of Der p 21 that are bound by antibodies, and with your findings.... could some extracts be hidden by the dimmers of Der p 21?

Additional comments

Interesting work. Please make it a bit better.

---

## Round 0.2 · accepted · Accept

The manuscript is accepted. The author's response to reviewers involved conducting a new experiment to validate the analysis and results. The authors also improved the discussion sections and declared all limitations and implications of the results.

·

Basic reporting

No Comment

Experimental design

No Comment

Validity of the findings

No Comment

Additional comments

No Comment